# Perspectives in Amyotrophic Lateral Sclerosis: Biomarkers, Omics, and Gene Therapy Informing Disease and Treatment

**DOI:** 10.3390/ijms26125671

**Published:** 2025-06-13

**Authors:** Nina Bono, Flaminia Fruzzetti, Giorgia Farinazzo, Gabriele Candiani, Stefania Marcuzzo

**Affiliations:** 1genT_LΛB, Department of Chemistry, Materials and Chemical Engineering “Giulio Natta”, Politecnico di Milano, 20131 Milan, Italy; nina.bono@polimi.it (N.B.); flaminia.fruzzetti@polimi.it (F.F.); gabriele.candiani@polimi.it (G.C.); 2Neurology 4—Neuroimmunology and Neuromuscular Diseases, Fondazione IRCCS Istituto Neurologico Carlo Besta, 20133 Milan, Italy; giorgia.farinazzo@istituto-besta.it; 3Brain-Targeted Nanotechnologies (BraiNs) Lab, Fondazione IRCCS Istituto Neurologico Carlo Besta—Politecnico di Milano, 20133 Milan, Italy

**Keywords:** amyotrophic lateral sclerosis, advanced diagnosis, therapeutic strategies, systems biology approach, bioengineering, nanotechnology

## Abstract

Amyotrophic lateral sclerosis (ALS) is a fatal neurodegenerative disease characterized by the progressive loss of upper and lower motor neurons, leading to muscle weakness, paralysis, and ultimately respiratory failure. Despite advances in understanding its genetic basis, particularly mutations in *Chromosome 9 Open Reading Frame 72* (*C9orf72*), *superoxide dismutase 1* (*SOD1*), *TAR DNA-binding protein* (*TARDBP*), and *Fused in Sarcoma* (*FUS*) gene, current diagnostic methods result in delayed intervention, and available treatments offer only modest benefits. This review examines innovative approaches transforming ALS research and clinical management. We explore emerging biomarkers, including the fluid-based markers such as neurofilament light chain, exosomes, and microRNAs in biological fluids, alongside the non-fluid-based biomarkers, including neuroimaging and electrophysiological markers, for early diagnosis and patient stratification. The integration of multi-omics data reveals complex molecular mechanisms underlying ALS heterogeneity, potentially identifying novel therapeutic targets. We highlight current gene therapy strategies, including antisense oligonucleotides (ASOs), RNA interference (RNAi), and CRISPR/Cas9 gene editing systems, alongside advanced delivery methods for crossing the blood–brain barrier. By bridging molecular neuroscience with bioengineering, these technologies promise to revolutionize ALS diagnosis and treatment, advancing toward truly disease-modifying interventions for this previously intractable condition.

## 1. Introduction

Amyotrophic lateral sclerosis (ALS) is a progressive neurodegenerative disorder characterized by the selective loss of motor neurons, leading to severe muscle weakness, paralysis, and ultimately, respiratory failure [1]. Disease progression and survival in ALS can vary significantly across different subtypes, influenced by clinical features such as age and site of onset. Patients with bulbar-onset ALS, who show early and selective motor neuron degeneration in the brainstem region, and older patients generally have a poorer prognosis [2]. ALS can occur in two primary forms: sporadic ALS (sALS), which accounts for approximately 90% of cases, and familial ALS (fALS). fALS can follow various inheritance patterns, including autosomal dominant, autosomal recessive, and X-linked, or may occur in the absence of a clear family history [3]. A number of genes have been implicated in both fALS and sALS. The most commonly associated gene in inherited ALS is *Chromosome 9 Open Reading Frame 72* (*C9orf72*), part of the C9orf72–SMCR8 complex. Other notable genes include encoding *TAR DNA-binding protein* (*TARDBP*), *superoxide dismutase 1* (*SOD1*), and *Fused in Sarcoma* (*FUS*) gene. Despite significant advancements in understanding its genetic and molecular underpinnings, ALS remains an incurable disease with limited therapeutic options. The heterogeneity of its pathophysiology [4], which involves oxidative stress, excitotoxicity, protein misfolding, neuroinflammation, and impaired RNA metabolism, poses major challenges for both diagnosis and treatment [5].

Traditional diagnostic approaches for ALS, which rely heavily on clinical observation and electro-physiological assessments [6,7], are often time-consuming and lack sensitivity in the early stages of the disease [8]. This frequently leads to substantial diagnostic delays, limiting the window for timely intervention and potentially compromising patient outcomes. Furthermore, the current therapeutic landscape, dominated by riluzole and edaravone [9], offers only marginal clinical benefits, primarily aimed at modestly slowing disease progression rather than halting or reversing it. These limitations strongly emphasize the critical and unmet need for innovative diagnostic tools and truly disease-modifying therapies that can improve prognosis and quality of life for ALS patients.

In recent years, ALS research has undergone a significant shift from a purely clinical perspective to a biology-driven approach, aiming to identify molecular signatures that stratify patients based on distinct pathophysiological mechanisms. Instead, emerging evidence supports the existence of multiple biologically distinct endotypes, each with specific molecular underpinnings and potential druggable targets [10]. In this evolving landscape, high-throughput omics technologies, including genomics, transcriptomics, epigenomics, proteomics, and metabolomics, have become essential tools for deepening our understanding of ALS and paving the way toward personalized diagnostics and therapies.

Multidisciplinary research efforts have focused on developing innovative diagnostic tools [11], identifying reliable biomarkers [12], and exploring targeted therapeutic approaches [13,14]. While previous reviews have examined these domains individually, few have explored their interconnected potential for advancing precision medicine.

The primary objective of this comprehensive review is to critically analyze and synthesize current evidence on how three key innovative approaches are transforming ALS research and clinical practice: (1) emerging biomarker discovery for early diagnosis and patient stratification, (2) multi-omics integration for understanding disease heterogeneity and identifying therapeutic targets, and (3) advanced gene therapy strategies for disease-modifying interventions. Specifically, this review aims to evaluate the clinical translation potential of these approaches, identify current limitations and future directions, and assess their collective impact on advancing toward personalized medicine in ALS. By integrating findings from molecular neuroscience, clinical research, and bioengineering, we seek to provide a roadmap for the next generation of ALS diagnostics and therapeutics.

## 2. Redefining ALS Diagnosis and Treatment Through Biomarker Discovery

Our analysis of current diagnostic methods reveals that traditional approaches, which rely primarily on clinical evaluation and electrophysiological assessments [4,5], are often insufficiently sensitive in the early stages of ALS. As a result, they frequently lead to significant delays in diagnosis, severely limiting opportunities for timely intervention. Moreover, these approaches offer little support in the discovery or validation of novel disease-modifying treatments, highlighting a critical gap in the current diagnostic framework. Therefore, a new approach to disease classification is urgently needed.

A biologically based classification system would enable earlier diagnosis and facilitate the identification of patient-specific cellular and molecular mechanisms, paving the way for more targeted and advanced therapeutic strategies. The identification of reliable biomarkers, as biological indicators such as DNA, RNA, or proteins that correlate with disease progression or treatment response, is critically important to improve ALS diagnosis, patient monitoring, and the development of effective therapies. Biomarkers can serve several different functions, thus improving care and research. They can be categorized as (Figure 1): (i) diagnostic biomarkers, (ii) prognostic biomarkers, and (iii) predictive biomarkers.

Diagnostic biomarkers help detect or confirm the presence of ALS, aiding early diagnosis and treatment [15]. These biomarkers could also potentially reveal and differentiate ALS subtypes since ALS is a clinically and pathophysiological heterogeneous condition [16].

On the other hand, prognostic biomarkers enable predicting the severity and progression of the disease, guiding treatment plans and clinical trials [17]. For clinical trials, these biomarkers would facilitate patient stratification based on differences in disease severity and progression.

Predictive biomarkers show biological changes after treatment exposure and help identify individuals likely to benefit from specific therapies, improving clinical trial designs [18].

In ALS, biomarkers would allow an earlier and more accurate diagnosis, with the opportunity to start an earlier treatment able to modify the disease course. They could help the classification/stratification of ALS patients, monitor the disease progression, and identify patients who will respond better to a particular drug. Biomarkers can also provide a valuable tool for the identification of new therapeutic approaches and drive patients’ enrollment in clinical trials.

Table 1 offers a concise summary of traditional and promising diagnostic, prognostic, and predictive methods in ALS.

### 2.1. Fluid-Based Biomarkers

Fluid-based biomarkers detection represents a minimally invasive approach to understanding disease pathophysiology and tracking progression in ALS. These biomarkers, primarily analyzed from cerebrospinal fluid (CSF) (collected through lumbar puncture) and blood (serum/plasma) (from simple venipuncture), offer valuable insights into the underlying pathological processes while avoiding the need for invasive tissue sampling [12,19].

#### 2.1.1. Neurofilament Light Chain

Among the fluid-based biomarkers, neurofilament light chain (NfL) analysis is considered one of the most established and widely used approaches in the clinical assessment of neurodegenerative diseases, including ALS. NfL is a cytoskeletal protein predominantly expressed in axons of myelinated neurons. When neurons undergo damage or degeneration, NfL is released into the CSF and subsequently diffuses into peripheral circulation, making it detectable in blood plasma. NfL levels significantly increase after symptom onset and typically stabilize within a year [20]. The ability to measure NfL in both CSF and blood has made it an increasingly accessible biomarker for monitoring ALS progression and potentially for diagnostic purposes. However, despite their value as prognostic indicators, NfL levels lack disease specificity for ALS diagnosis. Studies have demonstrated that CSF NfL levels are significantly increased not only in ALS but also in other neurological disorders such as Alzheimer’s disease, multiple sclerosis, and traumatic brain injury, reflecting NfL’s role as a general marker of neuronal damage rather than an ALS-specific biomarker [21]. NfL serves as a reliable indicator of neuronal injury, with its levels correlating closely with the extent and rate of neurodegeneration [12,21]. Higher baseline NfL concentrations in both CSF and blood are associated with greater disease severity, faster progression, shorter survival, and upper motor neuron degeneration. While useful for monitoring disease progression, the role of NfL in early detection remains limited, as evidence suggests that elevated levels may occur only in later, symptomatic stages, limiting its diagnostic potential [22].

#### 2.1.2. Exosomes

Exosomes, emerging as novel biomarkers in ALS, are a type of naturally occurring extracellular vesicles ranging from 30–150 nm in diameter and represent a promising new avenue for early and non-invasive disease diagnosis. Extracted from CSF or blood, exosomes contain multiple ALS-related biomarkers proteins (SOD1, TDP-43, pTDP-43, FUS), nucleic acids, lipids, or other molecules that can be measured [12,23,24]. Recent investigations have identified novel potential ALS biomarkers through exosome analysis, such as differentially expressed microRNAs (miRNAs) [25] and significant alterations in lipid composition profiles in plasma [26,27] and CSF [28]. Notably, TDP-43 and NfL extracted from plasma exosomes have shown increase in ALS patients, suggesting their potential prognostic role [24]. A particular advantage of exosome-based approaches lies in their ability to naturally cross the blood–brain barrier (BBB) and enable cell-type specificity through surface marker selection, offering a distinct advantage over conventional CSF or blood assays, which typically cannot determine the cellular origin of detected analytes [29,30,31]. This cellular specificity may enhance biomarker accuracy by focusing analysis on vesicles derived from disease-relevant cell populations.

#### 2.1.3. Non-Coding RNA

Non-coding RNAs (ncRNAs), particularly microRNAs (miRNAs)—endogenous small non-coding RNAs approximately 22 nucleotides in length that regulate gene expression through the same RNAi machinery—represent a promising frontier in ALS biomarker development and a novel method for identifying molecular signatures associated with disease onset and progression. These small, non-coding regulatory RNAs function at the post-transcriptional level to regulate gene expression [32], and have emerged as potential biomarkers for various neurological disorders due to their remarkable stability in both CSF and blood [33]. Recent technological advances in RNA sequencing have significantly improved the ability to generate high-quality libraries from small RNA samples, enabling the screening of ncRNA biomarkers from limited clinical specimens [34]. Several studies have already employed RNA sequencing to identify differentially expressed miRNAs in the CSF and blood of ALS patients that could serve as molecular diagnostic, prognostic, and/or predictive biomarkers [35,36]. Of particular interest, miR-146a has been shown to differentiate between patients with upper motor neuron predominance and those with lower motor neuron predominance. The study highlights the potential role of miR-146a in peripheral axon impairment and suggests its value as both a diagnostic and prognostic biomarker for ALS [16].

#### 2.1.4. Cryptic Peptides

The identification of ALS-specific biomarkers has focused on TDP43, a protein involved in both ALS and frontotemporal dementia (FTD). In these pathophysiologic conditions, TDP43 mislocalizes from the nucleus to the cytoplasm, disrupting RNA translation and causing the inclusion of abnormal exons, known as cryptic exons, which are usually excluded from mature mRNAs [28]. This mislocalization affects critical neuronal genes like *Unc-13 Homolog A* (*UNC13A*) and *stathmin-2* (*STMN2*) [37], leading to their reduced expression. The inclusion of cryptic exons results in the production of faulty transcripts that create truncated or novel proteins, which impair physiological cellular function. These cryptic peptides are valuable biomarkers as they are directly tied to TDP43 pathology. As such, they offer a promising diagnostic tool for ALS and may help classify patients into subphenotypes, aiding in more precise clinical management [29,30]. In this context, cryptic peptides represent an emerging biomarker class for ALS, with strong potential for advancing early and pathology-specific diagnosis.

#### 2.1.5. Neuroinflammation and Metabolism

ALS consists not only in neuronal death but also in a range of non-neuronal processes, including neuroinflammation and dysregulated metabolism [31,32], both of which provide valuable insights for biomarker development. Chitinase family proteins have garnered significant attention as biomarkers of neuroinflammation in ALS. Specifically, chitotriosidase (CHIT1), chitinase-3-like-1 (CHI3L1), and chitinase-3-like-2 (CHI3L2) are associated with faster disease progression, cognitive dysfunction, and shorter survival [33]. Similarly, the glial protein S100 Calcium Binding Protein B (S100B) has prognostic value, with its lower CSF concentration correlating with better survival outcomes [34]. Furthermore, a reduction in FOXP3 and CD25-positive regulatory T cells (T_reg_), anti-inflammatory immune T cells that protect motor neurons and maintain an immunologically tolerant environment, has been linked to ALS progression. As the disease advances, the ratio of activated-to-resting T_reg_ cells decreases [36]. These findings have contributed to the development of immunomodulatory therapies for ALS, with the proportion of circulating T_reg_ cells now serving as a biomarker to monitor treatment efficacy.

In addition to CSF biomarkers, urine offers a less invasive alternative for monitoring disease progression. Elevated levels of p75 and neopterin in urine are observed in ALS patients and increase as the disease progresses, indicating the potential of these molecules to predict responses to anti-inflammatory therapies [35,37].

Together, these neuroinflammatory biomarkers provide a multifaceted approach to monitoring ALS progression and evaluating therapeutic efficacy.

**Table 1 ijms-26-05671-t001:** Traditional and promising diagnostic, prognostic, and predictive methods in ALS.

	Details	Roles	References
**Traditional diagnostic and prognostic methods**			
Clinical evaluation	El Escorial criteria; Awaji criteria; Gold Coast criteria; The King’s Clinical Staging System and the Milano–Torino (MiToS) Functional Staging System; ALS Functional Rating Scale—Revised (ALSFRS-R).	Diagnostic evaluation	[5]
Electrophysiological assessments	Muscle action potential.	Diagnostic evaluation	[6]
Neurofilament light chain (NfL) analysis	NfL is a sensitive but non-specific biomarker of neuronal damage whose elevated levels in CSF and blood correlate with ALS progression and severity.	Diagnostic and prognostic fluid biomarkers	[20,21,22]
**Promising non-invasive diagnostic, prognostic, and predictive methods**			
Exosomes analysis	Exosomes are extracellular vesicles capable of crossing the blood–brain barrier and carrying ALS-related biomarkers (e.g., TDP-43, NfL, miRNAs), offering a promising, non-invasive, and cell-specific approach for early diagnosis and disease monitoring.	Prognostic and predictive fluid biomarkers	[23,24,25,26,27,28,29,30,31]
Non-coding RNA profiling	miRNAs, small non-coding RNAs involved in gene regulation expressed in CSF, blood, and serum, represent a promising avenue for identifying molecular signatures linked to ALS onset and progression.	Diagnostic, prognostic, and predictive fluid biomarkers	[16,33,34,35,36]
Cryptic peptide analysis	Cryptic peptides, closely linked to TDP-43 pathology, are emerging biomarkers expressed in CFS, plasma, and serum, with strong potential for early, pathology-specific ALS diagnosis and patient subtyping.	Diagnostic and predictive fluid biomarkers	[30,31,32,37]
Neuroinflammation and metabolism investigation	Chitinase family, specifically CHIT1, CHI3L1, and CHI3L2 expressed in CSF;	Prognostic and predictive fluid biomarkers	[35]
	S100B expressed in CFS;		[36]
	T_reg_ cells expressed in CSF;		[38]
	p75 and neopterin expressed in urine.		[37,38]
Neuroimaging and electrophysiological assessment	MRI to measure brain, spinal cord, and muscle volume;	Diagnostic and prognostic non-fluid biomarkers	[39,40,41,42,43,44,45]
	EMG to measure fasciculations early.		[46,47]
	PET to measure metabolic changes.		[48,49,50,51]

### 2.2. Non-Fluid Biomarkers in ALS

Beyond fluid-based biomarkers, ALS research increasingly relies on non-fluid biomarkers, objective measurable indicators that do not require sampling of blood, CSF, or other bodily fluids. These non-fluid biomarkers are classified into three major categories: neuroimaging-based, immunohistochemistry-based, and electrophysiological biomarkers, each providing distinct insights into disease mechanisms and progression [38].

Neuroimaging biomarkers provide direct measurements of structural and functional changes in the nervous system affected by ALS. They include molecular, functional, and structural measurements that can be derived from various technologies including Magnetic Resonance Imaging (MRI), Positron Emission Tomography (PET), and electrophysiological recordings (Table 1 and Figure 1).

#### 2.2.1. Magnetic Resonance Imaging

Structural and functional neuroimaging techniques have faced challenges, such as inconsistent results and varying accuracy [52], but some studies indicate that brain atrophy measured by changes in brain volume and tissue loss by MRI may be linked to ALS progression, suggesting it could have prognostic value [39]. Larger studies are needed to confirm these findings, and improvements in imaging technology will enhance their use in both clinical and research settings.

Computational neuroimaging provides valuable insights into ALS progression and disease patterns. Efforts to standardize imaging protocols at national [40] and international [41] levels have improved sample sizes and validated findings. Longitudinal protocols with short follow-up periods have been crucial in mapping disease progression. New imaging technologies, like advanced scanners and improved spinal cord protocols, which measure the spinal cord volume, cross-sectional area, changes in tissue integrity, atrophy, and demyelination, allow for faster, more effective tracking of ALS, even in its pre-symptomatic phase. High-resolution MRI platforms, such as 3T and 7T, have deepened our understanding of ALS by allowing for a better assessment of disease spread and progression. Multimodal imaging techniques, using volumetric, morphometric, vertex, and cortical thickness analyses through a Bayesian approach permitted to segment the thalamus, amygdala, and hippocampus into specific nuclei and anatomically defined subfields, and helped evaluate the integrity of neural networks, and machine learning methods have identified subgroups of ALS patients based on clinical, radiological, and genetic features [42]. Imaging also helps detect early signs of ALS in individuals with genetic variants like *SOD1* and *C9orf72*, even before symptoms emerge [43,44,45].

#### 2.2.2. Electromyography and Positron Emission Tomography

Muscle imaging, including MRI and ultrasound, complements brain imaging in ALS diagnosis. Ultrasound, in particular, helps detect fasciculations early and is more sensitive than electromyography (EMG) [46,47]. Combining muscle ultrasound with EMG could identify lower motor neuron dysfunction earlier, improving diagnosis and clinical trial inclusion.

PET imaging is widely used to detect metabolic changes in ALS patients, both symptomatic and pre-symptomatic. It can also track therapy responses at the cellular level and reveal metabolic signatures linked to ALS phenotypes and genotypes [48,49]. Recent studies combining PET and MRI have improved data analysis and anatomical precision. Additionally, quantitative susceptibility mapping is becoming a valuable tool in imaging protocols for diagnosis and prognosis [50,51]. Machine learning and artificial intelligence (AI) are being increasingly used in ALS research, especially for analyzing PET and MRI data. While these technologies show diagnostic and prognostic potential, the lack of large-scale studies is a barrier to their widespread application [53]. One study demonstrated AI’s ability to stratify ALS patients into different phenotypes based on MRI data [42]. Future studies with larger sample sizes are needed to unlock the full potential of AI and machine learning in ALS biomarker discovery.

ALS is a complex, heterogeneous disease with varying pathogenic processes across patient groups. Biomarkers are needed for diagnosis, prognosis, and tracking disease progression. As no single biomarker can capture all aspects of ALS, a multimodal approach combining clinical data with diagnostic, prognostic, and pharmacodynamic markers is necessary. This may involve clinical evaluations, biological-fluid biomarkers, imaging, and neurophysiological tests.

Overall, findings from biomarker research demonstrate that both fluid-based and non-fluid biomarkers offer unique but complementary insights into ALS pathophysiology, with each category addressing different aspects of disease progression and diagnosis. While individual biomarkers like NfL and neuroimaging measures show promise, the complexity and heterogeneity of ALS necessitate integrated multi-modal approaches that combine fluid biomarkers, advanced neuroimaging, and electrophysiological assessments. Future biomarker strategies should focus on developing comprehensive panels that leverage the strengths of both fluid and non-fluid approaches to achieve the diagnostic precision and patient stratification capabilities required for personalized ALS management.

## 3. Understanding ALS Pathophysiology: Multi-Omics Integration

Research on ALS has increasingly shifted from a predominantly clinical and descriptive approach toward a biologically driven paradigm. This transition reflects the growing need to identify molecular signatures capable of stratifying patients based on underlying pathophysiological mechanisms, with the ultimate aim of guiding the design of personalized clinical trials and targeted therapies.

In this context, the integration of high-throughput “omics” approaches, spanning genomics, transcriptomics, epigenomics, proteomics, and metabolomics, has significantly transformed our understanding of ALS. These technologies enable the quantification of genes, mRNAs, proteins, and metabolites, providing insights into the complex interactomes of biological systems [54]. This progress has enhanced our comprehension of ALS’s molecular architecture, allowing the identification of distinct patient subtypes, and laiding the groundwork for discovering biomarkers and developing individualized treatments. Given the multifactorial nature of ALS, the integration of various omics layers is crucial for a holistic understanding of the disease, opening new avenues for advanced diagnostics and the development of innovative therapies. Figure 2 illustrates the main omics approaches and their integration to uncover the molecular mechanisms underlying ALS pathophysiology and identify novel therapeutic targets for innovative treatments.

### 3.1. Genomics

Genomic studies have identified several genetic mutations associated with ALS, including well-known genes such as *SOD1*, *C9orf72*, *FUS*, and *TARDBP* [55]. Understanding these genetic factors helps to define fALS and provides insights into disease pathways and potential therapeutic targets. Advances in genome-wide association studies and sequencing technologies, such as whole-genome and whole-exome sequencing, have led to the identification of additional causative and susceptibility genes or rare genetic variants that could also play a role in sporadic ALS [56,57]. The identified genes are involved in critical processes like cytoskeleton remodeling, mitochondrial metabolism, autophagy, RNA processing, and DNA repair [13]. Genomic analysis also helps explain differences in prognostic profiles among ALS patients. Certain mutations are associated with longer survival, such as loss-of-function mutations in *Ephrin Type-A Receptor 4* (*EPHA4*) gene, while others like those in the *SOD1* gene can impact disease progression [58,59]. In addition to point mutations, copy number variations (CNVs), including rare ALS-specific loci, have been identified, influencing gene expression and contributing to ALS pathogenesis [60].

### 3.2. Transcriptomics and Epigenomics

Transcriptomic studies play a crucial role in understanding the molecular mechanisms underlying ALS by analyzing gene expression profiles, including mRNA, long non-coding RNAs (lncRNAs), and miRNAs [61].

High-throughput RNA sequencing (RNA-seq) and microarray technologies have revealed dysregulated transcriptional networks associated with neurodegeneration, oxidative stress, RNA metabolism, and protein homeostasis [62]. Dysregulation of key RNA-binding proteins such as TDP-43, FUS, and heterogeneous nuclear ribonucleoproteins (hnRNPs) alters transcript stability, splicing, and transport, contributing to ALS pathogenesis [63]. Epigenetic modifications, including histone modifications and DNA methylation, further influence transcriptomic changes, which could provide insights into disease pathophysiology and help the identification of potential pharmacological targets. Hop et al. [64] analyzed DNA methylation patterns in almost 10,000 individuals with ALS and healthy controls and identified 45 differentially methylated positions (DMPs) annotated to 42 genes. The changes involved genes associated to metabolic and inflammatory pathways, and the authors identified several DMPs associated with disease progression in their cohort. The results aid in identifying disease-relevant mechanisms that can be targeted to block or delay ALS progression [39]. In the context of C9orf72-ALS [65], N6-methyladenosine (m6A), the primary common internal mRNA modification, is reduced in C9orf72-ALS, both in patient-derived induced pluripotent stem cell (iPSC)-differentiated neurons, and postmortem brain tissues. This global m6A hypomethylation leads to widespread mRNA stabilization and increased gene expression, particularly in pathways related to synaptic activity and neuronal function, contributing to disease progression [66]. Additionally, a relevant role in the pathogenies of ALS was attributed to muscle-specific miRNAs (myomiRs), a group of miRNAs that play a crucial role in maintaining muscle homeostasis [67]. MyomiRs can modulate their expression in response to environmental factors like exercise, as well as genetically determined changes [68]. Given their role in muscle-motor neuron communication, myomiRs are strong candidates for ALS studies. They regulate genes involved in myogenesis [67], potentially influencing muscle atrophy pathways and neuromuscular junction consolidation in ALS. Single-cell transcriptomics is emerging as a powerful tool to dissect cellular heterogeneity within the central nervous system (CNS), providing insights into glial activation, neuronal degeneration, and potential therapeutic targets [69]. Single-nucleus transcriptomics and epigenomics in post-mortem motor cortices from C9orf72-ALS showed pervasive alterations of gene expression with concordant changes in chromatin accessibility and histone modifications. The greatest alterations occur in upper and deep layer excitatory neurons, as well as in astrocytes. In neurons, the changes imply an increase in proteostasis, metabolism, and protein expression pathways, alongside a decrease in neuronal function. In astrocytes, the alterations suggest activation and structural remodeling. These findings highlight a context-dependent molecular disruption in C9orf72-ALS, indicating unique effects across cell types, brain regions, and diseases [70]. Overall, transcriptomic studies in ALS are advancing our understanding of disease mechanisms, aiding in biomarker discovery, and offering new avenues for therapeutic interventions, particularly in RNA-targeted therapies such as antisense oligonucleotides (ASOs) and small interfering RNAs (siRNAs).

### 3.3. Proteomics

Proteomic profiling has revealed significant changes in protein expression and post-translational modifications in ALS patients, offering potential biomarkers and drug targets. In ALS and FTD, two neurodegenerative diseases with overlapping clinical, neuropathological, and genetic features, proteomic analysis is essential for exploring their molecular connections. Notably, among the many proteins identified, both ALS and FTD brain and spinal cord samples show similar disruptions in mitochondrial and metabolic pathways, highlighting shared dysfunctions between the two diseases [71]. Similar analyses of brain tissues have provided valuable insights into the different subtypes of ALS and FTD, distinguishing them based on variations in protein levels, as well as differences in the assembly, distribution, and morphology of protein aggregates [72,73]. An analysis of the CSF proteome in ALS patients identified a panel of candidate biomarkers associated with synaptic activity, inflammation, glial response, axonal damage, and apoptosis [13,74,75]. Additionally, proteomic analysis of CSF-derived extracellular vesicles, aimed at providing novel insights into key ALS pathogenesis processes, revealed downregulation of proteasome core complex proteins through gene ontology enrichment analysis. This finding suggests a potential impairment in protein degradation pathways, which could contribute to neurodegeneration and serve as a promising target for therapeutic intervention [76].

### 3.4. Metabolomics

Metabolomics has great potential in personalized medicine, enabling the identification of key biological markers for diagnosis, disease monitoring, and the discovery of novel pathways and therapeutic targets. In ALS, metabolomic analysis provides valuable insights into disrupted metabolic processes and energy imbalances linked to neuronal dysfunction and death [77]. By analyzing metabolites such as glutamate, antioxidants, and lipids, insights have been gained into mechanisms like glutamatergic excitotoxicity, oxidative stress, and mitochondrial dysfunction, all of which are involved in ALS etiology [13]. Metabolomics in plasma samples from ALS patients and controls has confirmed key factors in ALS etiology, such as sphingolipids, which regulate autophagy and inflammation. Additionally, it has uncovered novel metabolic abnormalities, including disruptions in benzoate metabolism, potentially linked to pesticide exposure, and diacylglycerols, which are involved in inflammation, immune signaling, and apoptosis [78]. Changes in specific metabolites associated with energy deficit and neurotoxic effects have been identified in both peripheral blood and CSF of ALS patients, making them potential biomarkers or therapeutic targets [79]. Additionally, basal tear metabolomics does not represent a diagnostic tool discriminating patients with ALS and controls and does not correlate with disease severity or evolution, but discriminates patients with bulbar and spinal forms of ALS, representing an innovative method to better characterize ALS phenotypic heterogeneity [80].

### 3.5. Multi-Omics Integration in ALS

Individual omics layers (genomics, transcriptomics, proteomics, metabolomics) each provide valuable but incomplete insights into ALS pathophysiology, with true mechanistic understanding emerging only through their integration. While significant advances have been made in single-omics approaches, the heterogeneous nature of ALS requires comprehensive multi-omics frameworks that can capture the complex interactions between genetic variants, gene expression patterns, protein alterations, and metabolic disruptions [54]. Integrating multiple layers of biological data enables the identification of intricate interactions between molecular pathways, the classification of disease subtypes, and the detection of potential biomarkers for diagnosis, prognosis, and therapeutic response. This exclusive approach supports the development of targeted therapies, ultimately enhancing disease modeling and patient stratification.

Caldi Gomes et al. [14] performed a comprehensive multi-omics approach to investigate the early and sex-specific molecular mechanisms underlying ALS. The prefrontal cortex of 51 patients with sporadic ALS and 50 control subjects was analyzed, revealing significant molecular alterations associated with the disease. Males exhibited more pronounced changes in molecular pathways compared to females. An integrated analysis of transcriptomes, (phospho) proteomes, and miRNAomes identified distinct ALS subclusters, characterized by differences in immune response, extracellular matrix composition, mitochondrial function, and RNA processing. The mitogen-activated protein kinase (MAPK) pathway was highlighted as an early disease mechanism. Furthermore, trametinib, a MAPK inhibitor, demonstrated potential therapeutic benefits in vitro and in vivo, particularly in females, suggesting a promising direction for targeted ALS treatments [14].

A recent study introduced a network medicine approach that integrates multi-omics data from the human brain to prioritize drug targets and potential repurposable treatments for ALS. The research focused on the impact of non-coding ALS loci identified in genome-wide association studies on various brain-related quantitative trait loci (QTL), including expression (eQTL), protein (pQTL), splicing (sQTL), methylation (meQTL), and histone acetylation (haQTL). Using a network-based deep learning framework, the authors identified 105 ALS-associated genes, which were enriched in established ALS-related pathways. By applying network proximity analysis to connect these genes with drug-target networks within the human protein–protein interactome model, they identified potential repurposable drugs, such as diazoxide and gefitinib, for ALS. This network-based multi-omics framework offers a powerful tool for identifying drug targets and potential treatments for ALS [81].

It is worth of note that integrated multi-omics data in the field of ALS are being deposited into public repositories to accelerate research progress. A leading example is Answer ALS (https://www.answerals.org/) (accessed on 30 May 2025), recognized as the world’s largest and most comprehensive ALS research initiative, which emerged from the understanding that isolated research efforts and simple animal models were insufficient to tackle the disease’s complexity. This resource comprises extensive biological and clinical data from over 1000 ALS patients and healthy controls, creating an unprecedented opportunity to identify clinical-molecular-biochemical ALS subtypes through population-level analysis. The project’s core strength lies in its multi-omics approach, where human iPSCs are differentiated into spinal neurons that undergo comprehensive multi-omics profiling, including whole-genome sequencing to identify genetic variants, RNA transcriptomics to assess gene expression patterns, Assay for Transposase-Accessible Chromatin using sequencing (ATAC-seq) to evaluate chromatin accessibility, and proteomics to measure protein abundances.

Through sophisticated bioinformatics and computational analysis, these integrated datasets support the generation of clinical and biological signature that can uncover disease mechanisms and subgroups. To maximize scientific impact, all data are made accessible through an open-access web portal that enables researchers worldwide to perform their own analyses. This collaborative approach employs AI and machine learning to identify potential therapeutic targets, biomarkers for diagnosis and disease monitoring, and strategies for patient stratification in clinical trials [82].

In the near future, research should prioritize the development of standardized multi-omics integration platforms and computational tools that can translate these molecular signatures into clinically actionable biomarkers and therapeutic targets for precision medicine approaches.

## 4. The Evolution of ALS Treatment: From Current Options and Innovative Approaches

### 4.1. Current Treatment Landscape and Limitations

The development of effective therapies for ALS remains a significant challenge in modern medicine. Numerous clinical trials are investigating various therapeutic approaches targeting different pathological mechanisms of ALS. For instance, gene therapy approaches offer the potential to correct or silence disease-causing mutations and deliver neuroprotective factors directly to affected tissues. Stem cell therapies aim to replace lost motor neurons or provide trophic support to prevent further neurodegeneration. Approaches targeting protein misfolding and aggregation address a key pathological mechanism common across ALS subtypes.

These range from ASOs designed to silence mutated genes, to adeno-associated virus (AAV)-delivered gene therapies, stem cell treatments, and small molecule interventions. The diversity of approaches reflects both the complex multifactorial nature of ALS and the rapidly evolving landscape of therapeutic technologies [83]. Table 2 summarizes the most significant ongoing and recently completed clinical trials for ALS, highlighting their mechanisms, development stages, and administration routes. For a more comprehensive list of ongoing trials, refer to ClinicalTrials.gov and recent reviews on ALS clinical development [84,85,86].

Despite numerous clinical trials conducted over the last two decades, the majority have failed to show promising outcomes. Of more than 80 human clinical trials completed, indeed, only three medications have received U.S. Food and Drug Administration (FDA) approval, all offering modest benefits in terms of slowing disease progression by targeting non-specific factors such as excitotoxicity or oxidative stress [87]. Riluzole (brand names: Rilutek, Tiglutik), the first approved treatment (1995), extends survival by merely 2–3 months through glutamate antagonism [88]. Edaravone, approved in 2017, is a free radical scavenger that moderately slows functional decline in specific patient subsets but requires daily intravenous administration regimens [89]. The recently approved AMX0035 (Relyvrio; 2022) targets endoplasmic reticulum stress and mitochondrial dysfunction with encouraging but still incremental survival benefits [90].

These approved treatments share fundamental limitations: they address single pathways in a disease characterized by multiple interacting mechanisms, fail to target specific genetic causes, cannot restore lost motor neurons, and face significant challenges in crossing the BBB to achieve adequate therapeutic concentrations in the CNS. Perhaps most significantly, they provide symptomatic relief rather than disease modification, highlighting the urgent need for transformative approaches toward a true disease-modifying treatment for this devastating illness.

The limited success of ALS clinical trials can be ascribed to multiple factors, with a poor understanding of disease mechanisms being paramount. Additional challenges include limited bioavailability of therapeutic agents, inefficient therapeutic delivery to the CNS, difficulties in administration, lack of effective biomarkers for patient stratification and treatment response monitoring, late diagnosis reducing the therapeutic window, and lack of clinically relevant disease models that fully recapitulate human pathophysiology. The multifactorial nature of ALS and the complexity of its pathogenic mechanisms have made it exceptionally difficult to develop interventions that effectively address the disease progression.

Recent advances in the understanding of ALS pathophysiology, including the discovery of many of its genetic underpinnings, have enabled the development of targeted therapies, with today’s clinical trials holding growing promise for more efficacious treatments that address the fundamental limitations of current therapeutic approaches.

Numerous clinical trials are currently investigating various therapeutic approaches targeting different pathological mechanisms of ALS. These range from ASOs, designed to silence mutated genes, to AAV-delivered gene therapies, stem cell treatments, and small molecule interventions. The diversity of approaches reflects both the complex multifactorial nature of ALS and the rapidly evolving landscape of therapeutic technologies [3].

**Table 2 ijms-26-05671-t002:** ALS gene therapy in clinical trials.

Name	Drug Type	Target	Delivery System	Phase	Clinical Trial	Developer	Year (Start)	Route of Administration	Key Information
Tofersen (BIIB067)	ASO	*SOD1* mutations	Modified oligonucleotide backbone	Phase 3 completed	NCT02623699	Biogen	2018	Intrathecal	Received FDA accelerated approval in 2023. VALOR study showed slowing of disease progression in SOD1-ALS patients.
BIIB078	ASO	*C9orf72* mutations	Modified oligonucleotide backbone	Phase 1/2	NCT03626012	Biogen	2019	Intrathecal	It targets the most common genetic cause of ALS.
ION363 (Jacifusen)	ASO	*FUS* mutations	Modified oligonucleotide backbone	Phase 3	NCT04768972	Ionis Pharmaceuticals	2020	Intrathecal	Specifically for patients with *FUS* mutations. Named after Jaci Hermstad, an ALS patient.
VM202 (Engensis)	Plasmid DNA	*HGF* expression	Non-viral plasmid vector	Phase 2	NCT02427464	Helixmith	2018	Intramuscular	Non-viral plasmid DNA containing the *HGF* gene, enabling cells to produce this protein to promote nerve regeneration.
AAV-GDNF	*GDNF* gene	Neuroprotection	AAV9 viral vector	Phase 1/2	NCT01621581	Various institutions	2019	Intrathecal	It delivers the gene for GDNF to promote production of this neuroprotective protein by transduced cells. One-time administration.
AL001	Autologous T_reg_ cell therapy	Immune regulation	AAV9 viral vector	Phase 1/2	NCT05053035	Asklepios BioPharmaceutical	2020	Intrathecal	One-time AAV delivery of the hepatocyte growth factor gene, allowing transduced cells to express this neuroprotective protein.
AMX0035 (Relyvrio)	Small molecule combination (sodium phenylbutyrate and taurursodiol)	Cellular death pathways	-	FDA Approved	NCT03127514	Amylyx Pharmaceuticals	2017	Oral	Combination therapy. FDA approved in 2022.
NurOwn	Mesenchymal stem cell therapy	Multiple neuroprotective pathways	Autologous MSCs	Phase 3 completed	NCT03280056	BrainStorm Cell Therapeutics	2017	Intrathecal	Autologous MSCs secreting neurotrophic factors. Mixed results in Phase 3.
Masitinib	Small molecule (tyrosine kinase inhibitor)	Mast cells, microglia	-	Phase 3	NCT03127267	AB Science	2017	Oral	Anti-inflammatory and neuroprotective effects.
MN-166 (Ibudilast)	Small molecule (PDE4 inhibitor)	Neuroinflammation	-	Phase 2/3	NCT02714036	MediciNova	2018	Oral	Anti-inflammatory and neuroprotective.
Ravulizumab (Ultomiris)	Monoclonal antibody (C5 complement inhibitor)	Complement system	-	Phase 3	NCT04248465	Alexion Pharmaceuticals	2019	Intravenous	It targets complement-mediated neuroinflammation.
AP-101	Recombinant human FGF-1 protein	Neuronal survival	-	Phase 3	NCT05039099	Artielle Pharmaceuticals	2019	Intrathecal	It promotes motor neuron survival.
AT-1501	Monoclonal antibody	CD40L	-	Phase 2	NCT04322149	Eledon Pharmaceuticals	2020	Intravenous	It targets neuroinflammation.
SBT-272	Small molecule (mitochondria-targeted peptide)	Mitochondrial dysfunction	-	Phase 1	NCT02297035	Stealth BioTherapeutics	2021	Oral	It improves mitochondrial function in neurons.

### 4.2. Emerging Gene Therapies

In response to the limitations of current ALS treatments, gene therapy has emerged as a promising strategy for developing truly disease-modifying interventions. Unlike conventional pharmacological approaches that primarily address symptoms, gene therapy targets the genetic underpinnings of the disease, offering several distinct advantages: potential single-administration durability, precise targeting of disease-causing mutations, direct delivery of neuroprotective factors, and cell-specific interventions [91,92].

The acceleration of gene therapy development for ALS has been driven by significant advances in our understanding of the genetic basis of the disease. The identification of causative genes such as *SOD1* [93], *C9orf72* [94], *FUS* [95], and *TARDBP* [96], has provided specific targets for therapeutic intervention. Today, more than 30 genes are associated with fALS [97,98,99,100], with major genetic causes summarized in Table 3.

Gene therapy for ALS encompasses several distinct approaches: (i) gene replacement, which introduces functional copies of dysfunctional genes [101]; (ii) gene augmentation, which delivers trophic factors and other disease-modifying genes [102]; (iii) gene silencing, which suppresses harmful gene expression using ASOs or RNA interference (RNAi) [103]; (iv) gene editing, which directly corrects genetic mutations using technologies such as CRISPR/Cas9 [104,105].

#### 4.2.1. Gene Replacement and Augmentation

Gene replacement and gene augmentation approaches are particularly relevant for loss-of-function mutations but can also provide therapeutic benefits through the delivery of neuroprotective factors even in the absence of specific genetic targets.

Several clinical trials have explored gene augmentation strategies for ALS. VM202 (Engensis), a non-viral plasmid vector encoding hepatocyte growth factor (HGF), has been evaluated in Phase II trials with intramuscular administration. HGF has demonstrated neuroprotective and neuroregenerative properties in preclinical models of ALS [106].

For more efficient gene replacement and augmentation strategies, effective delivery systems are critical for transporting therapeutic genetic material to target cells. These vectors fall into two main categories: viral and non-viral, each with distinct characteristics [87,107].

Viral vectors, particularly AAVs and lentiviruses, are widely used due to their efficiency in gene transfer, ability to infect both dividing and non-dividing cells, and their potential for long-term gene expression [108]. AAVs are favored for ALS gene therapy for their safety profile, low immunogenicity, and stable transgene expression [105]. Different AAV serotypes exhibit varied tissue tropism, with AAV9 and AAV10 showing enhanced targeting towards the CNS, making them ideal candidates for treating neurodegenerative diseases like ALS. However, AAVs have limitations, including restricted DNA packaging capacity (approximately 4.7 kb), along with potential pre-existing immunity [87]. In this context, AAV-GDNF (glial cell-derived neurotrophic factor) has been investigated for its potential to promote motor neuron survival in ALS. This approach utilizes an AAV9 vector to deliver the *GDNF* gene intrathecally, enabling sustained production of this neuroprotective factor by transduced cells [35].

ALS001, an AAV9-delivered therapy developed by Asklepios BioPharmaceutical, represents another gene augmentation approach. This treatment delivers the HGF gene via a one-time intrathecal administration, allowing transduced cells to express this neuroprotective protein continuously. Phase I/II trials are evaluating its safety and preliminary efficacy (NCT05053035). Similarly, AAV-GDNF has been investigated for its potential to promote motor neuron survival in ALS. This approach utilizes an AAV9 vector to deliver the *GDNF* gene intrathecally, enabling sustained production of this neuroprotective factor by transduced cells [109].

While gene replacement strategies are theoretically appealing for monogenic forms of ALS, their implementation faces significant challenges. The large size of some ALS-associated genes exceeds the packaging capacity of AAV vectors, and achieving adequate expression levels in motor neurons throughout the CNS remains difficult. Lentiviral vectors offer advantages in DNA loading capacity (up to 8 kb) and sustained transgene expression. They can integrate into the host genome, providing persistent expression but also raising concerns about insertional mutagenesis [110]. Despite their utility, lentiviral vectors face obstacles such as limited diffusion from injection sites and relatively inefficient transduction of CNS cells compared to AAVs [111].

To address the limitations of viral systems, non-viral vectors like lipid nanoparticles (LNPs) and polymeric carriers are being explored as alternatives to address the limitations of viral systems. These approaches are still in the earliest stages of development for ALS therapy but present several potential advantages over viral vectors: significantly reduced immunogenicity, improved safety profiles, easier manufacturing, and the possibility of repeated administration—a critical advantage for treating chronic neurodegenerative diseases [107]. These characteristics open the door for novel treatment paradigms and clinical trial designs that would not be feasible with viral vectors due to pre-existing or treatment-induced immunity [107,110,112,113]. Despite these promising attributes, non-viral delivery systems currently face challenges of lower transfection efficiency and more transient gene expression compared to viral vectors, issues that remain the focus of ongoing research efforts.

Furthermore, these advances are creating opportunities for precision and personalized medicine approaches that account for individual genetic variability, moving from theoretical concepts toward clinical reality [114].

#### 4.2.2. Gene Silencing Approaches

ASOs. ASOs represent one of the most clinically advanced gene therapy approaches for ALS. These short, synthetic strands of DNA or RNA, typically 10 to 25 bases long, selectively bind to target mRNA sequences within cells [87]. By forming RNA–DNA or RNA–RNA hybrids, ASOs can modulate gene expression through several mechanisms, including RNase H-mediated mRNA degradation, inhibition of translation, or modulation of RNA splicing [115]. Chemical modifications—such as phosphorothioate backbones and 2′-O-methyl sugar groups—have significantly improved ASO stability, specificity, and resistance to nucleases, allowing for enhanced pharmacological activity and broader potential for clinical use [116].

One key advantage of ASOs is that they do not require viral vectors for delivery. Instead, they can be administered directly, typically via intrathecal injection for CNS disorders. While ASOs do not naturally cross the BBB, they can achieve widespread CNS distribution when delivered intrathecally, as demonstrated in both animal models and human trials [105]. However, challenges remain, particularly the need for repeated intrathecal administration and optimizing delivery efficiency [117].

In ALS, ASOs offer a promising therapeutic strategy by directly targeting the genetic drivers of the disease. One of the most studied targets is the *SOD1* gene, mutations of which cause toxic protein accumulation in motor neurons [100]. Tofersen (BIIB067) (Table 3), a next-generation ASO targeting SOD1, has progressed through clinical development, demonstrating significant reductions of SOD1 levels in CSF and a favorable safety profile. However, the VALOR study of Phase III did not reach the primary clinical efficacy endpoint at 28 weeks, although subsequent analyses suggested potential benefits with more prolonged treatment, leading to its accelerated approval by the FDA in 2023 [118], with the expectation that clinical benefit would be confirmed in subsequent studies.

*C9orf72* represents another critical ASO target, as hexanucleotide repeat expansions in this gene constitute the most common genetic cause of both ALS and FTD [119]. ASOs designed to target repeat-containing RNA in these expansions produce toxic RNA foci and dipeptide repeat proteins (DPRs) through repeat-associated non-AUG (RAN) translation. BIIB078, an ASO targeting C9orf72-ALS expansions (Table 2), has demonstrated efficacy in preclinical models by reducing both toxic RNA foci and DPRs without affecting healthy C9orf72 protein levels. In Phase I clinical trials, BIIB078 showed dose-dependent reductions in poly(GP) DPRs in the CSF, indicating target engagement. While the ASO was generally well-tolerated, Biogen announced in 2022 that the program would not advance to Phase III due to an unfavorable risk–benefit profile observed in the Phase I/II study, highlighting the challenges in translating preclinical success to clinical outcomes [120].

Another promising ASO target is *ataxin-2* (*ATXN2*), which modulates TDP-43 proteinopathy—a pathological hallmark present in approximately 97% of ALS cases. Intermediate-length polyglutamine expansions in ATXN2 increase ALS risk, and preclinical studies have shown that ATXN2 suppression can significantly reduce TDP-43 toxicity. In mouse models, ASO-mediated reduction of ATXN2 has extended survival and improved motor function, even when administered after symptom onset. This therapeutic approach is particularly promising because it targets a disease-modifying pathway relevant to both familial and sporadic ALS cases. A Phase I clinical trial (ATLAS) evaluating an *ATXN2*-targeting ASO (ION541) in ALS patients was initiated in 2020 by Ionis Pharmaceuticals in collaboration with Biogen, with preliminary results indicating good safety and tolerability [121].

RNAi-based Approaches. RNA interference represents a major mechanism of post-transcriptional gene silencing. This conserved biological process utilizes small ncRNA molecules to regulate gene expression post-transcriptionally. Within this framework, several specific approaches have been developed for therapeutic applications in ALS.

siRNAs are double-stranded RNA molecules, typically 21–23 nucleotides in length, which exhibit perfect complementarity to their target mRNAs. When incorporated into the RNA-induced silencing complex (RISC), siRNAs direct the cleavage of complementary mRNA targets, leading to their degradation [117]. Therapeutic siRNAs are designed for highly specific targeting of disease-related genes.

Short hairpin RNAs (shRNAs) are RNA molecules that form hairpin structures and are processed intracellularly by the enzyme Dicer into siRNAs.

Endogenous miRNAs are small non-coding RNAs that control gene expression through the same RNAi machinery. However, unlike siRNAs, miRNAs typically exhibit partial complementarity to their targets and regulate multiple genes simultaneously [116].

Beyond traditional approaches targeting miRNAs as ALS biomarkers, innovative strategies involve miRNAs as therapeutic agents. Such technologies can be broadly classified into two main categories, i.e., miRNA inhibition and miRNA replacement or enhancement (using miRNA mimics) [122]. For instance, anti-miRNA-155 reduced neuroinflammation and prolonged survival in ALS-model mice, while miRNA-206 upregulation improved neuromuscular junction regeneration [123]. Artificial miRNAs delivered via AAV vectors have been found to successfully targeted genes such as *SOD1*, significantly improving motor neuron survival and disease progression. Moreover, the ability of these vectors to use tissue-specific promoters and co-expression systems enhances their therapeutic precision [124].

In the context of ALS, RNAi has emerged as a promising gene silencing strategy, particularly for fALS cases linked to mutations in the *SOD1* gene. By selectively reducing the production of toxic mutant SOD1 protein, RNAi-based therapies aim to slow neurodegeneration and disease progression. Preclinical studies using both synthetic siRNAs and viral vector-delivered miRNAs or shRNAs have demonstrated efficacy in ALS animal models [121]. More advanced strategies involve the use of AAV vectors, particularly AAV9, to deliver miRNAs systemically or directly into the CNS [125]. A notable example is UMMS-001, an AAV9-delivered artificial miRNA targeting SOD1, which has shown robust suppression of the mutant gene and survival benefits in SOD1G93A mice [124]. Encouraged by these results, early-phase clinical trials have begun, including intrathecal infusion of an AAV-miR-SOD1 construct in patients. While some adverse effects, such as transient meningoradiculitis, were observed in one patient, another tolerated the treatment well under immunosuppression, and both cases suggested biological activity with lower SOD1 levels [126]. Despite challenges such as limited siRNA stability, potential immune responses, and off-target effects, the sustained expression enabled by AAV delivery and continued refinement of RNAi scaffolds hold great potential for translating these approaches into viable therapies for ALS and other neurodegenerative diseases.

Despite challenges, such as delivery across the BBB, miRNA stability, and off-target effects, preclinical evidence strongly supports miRNAs as viable therapeutic agents. Recent success in miRNA-based therapies for other neuropathies bolsters the potential for clinical translation in ALS [127]. Consequently, miRNAs represent a multifaceted tool in the fight against ALS, with applications spanning from early detection to therapeutic intervention.

CRISPR Gene Editing. CRISPR/Cas9 technology represents a revolutionary approach to genome editing with significant potential for ALS treatment [128]. This system, derived from bacterial immune mechanisms, uses a guide RNA to direct the Cas9 nuclease to specific genetic sequences, introducing double-stranded breaks that are subsequently repaired by cellular machinery [129]. Compared to older genome-editing technologies like zinc-finger nucleases or transcription activator-like effector nucleases, CRISPR offers greater simplicity, adaptability, and cost-effectiveness [121].

Mutations in the *SOD1* gene have been the most extensively studied targets for CRISPR/Cas9 intervention. Several landmark studies have demonstrated the therapeutic potential of this approach. For instance, Gaj et al. [130] delivered CRISPR/Cas9 components via AAV9 vectors through the facial vein in SOD1-G93A mice, achieving approximately 30% reduction in mutant SOD1 protein in the spinal cord. This intervention preserved motor neuron function, delayed disease onset, and extended survival by 28–30 days, though efficacy was greatest when administered to pre-symptomatic animals. Building on this work, Chen et al. [131] used novel AAV-PHP variants to achieve more efficient CNS delivery, resulting in improved motor function and extended survival in SOD1-ALS mouse models. CRISPR/Cas9 approaches targeting C9orf72 have primarily focused on excising the expanded GGGGCC repeat region. Meijboom et al. [132] demonstrated the therapeutic potential of this approach by administering AAV9 vectors containing guide RNAs and Cas9 into the bilateral striatum of C9orf72-associated mice, which reduced RNA foci, poly-dipeptide proteins, and ameliorated disease phenotypes. López-Gonzalez et al. [133] provided an in vitro proof of principle for this approach in iPSC-derived motor neurons from ALS patients, showing that CRISPR/Cas9-mediated excision of the pathogenic repeat expansion reduced toxic RNA foci and restored normal cellular phenotypes.

CRISPR/Cas9 correction of FUS and TDP-43 mutations has been primarily explored in patient-derived iPSCs, yielding important insights into disease mechanisms and therapeutic potential. Bhinge et al. [134] corrected the recessive H517Q mutation in the *FUS* gene using CRISPR/Cas9 and identified aberrant activation of ERK and p38 MAPK pathways as key mechanisms in FUS-related neurodegeneration. Guo et al. [135] demonstrated that CRISPR/Cas9-mediated genetic correction of *FUS* mutations (R521H) in patient-derived iPSCs rescued the cytoplasmic hyperexcitability and progressive axonal transport defects characteristic of FUS-ALS, establishing a causative relationship between these mutations and the observed phenotypes. Tann et al. [136] used CRISPR/Cas9 to correct the M337V mutation in *TARDBP* gene in iPSCs from ALS patients. When differentiated into excitatory cortical/hippocampal-like neurons, these corrected cells showed normalization of BDNF secretion, which is essential for neuron survival, differentiation, and synaptic plasticity. This study identified impaired BDNF function as a potential mechanism linking TDP-43 mutations to neurodegeneration.

Dafinca et al. [137] further extended this work by showing that CRISPR correction of TARDBP mutations reversed mitochondrial calcium uptake defects in patient-derived motor neurons, protecting against glutamate excitotoxicity. These cellular models provide a valuable proof of concept for *TARDBP*-targeted CRISPR interventions, though in vivo applications remain to be established.

Despite these significant technical and biological hurdles, CRISPR/Cas9 approaches offer distinct advantages over other gene therapy modalities, including greater targeting precision, versatility across multiple genetic drivers of ALS, and potential for permanent modification with a single intervention. As delivery technologies improve and newer CRISPR systems with enhanced safety profiles emerge, this revolutionary technology may ultimately provide the long-sought disease-modifying treatments for this devastating neurodegenerative disorder.

Key findings from gene therapy development reveal that while ASOs have achieved clinical success for specific genetic targets like *SOD1*, current approaches address only individual genetic causes in a disease characterized by multiple pathogenic mechanisms. CRISPR/Cas9, RNAi, and viral delivery technologies each show distinct advantages but also face unique limitations in terms of delivery efficiency, safety profiles, and therapeutic durability. Future gene therapy strategies should focus on developing combinatorial approaches that can simultaneously target multiple disease pathways, improving delivery systems for enhanced CNS penetration, and creating treatment protocols that can be adapted to different genetic and molecular subtypes of ALS to maximize therapeutic efficacy across the heterogeneous patient population.

**Table 3 ijms-26-05671-t003:** Major genetic causes of ALS.

Gene	Chromosome	Year Discovered	Protein Function	% of fALS	% of sALS	Key Features	References
*C9orf72*	9p21.2	2011	Membrane trafficking; autophagy	30–40%	5–10%	Hexanucleotide (GGGGCC) repeat expansion; most common genetic cause of ALS/FTD; complex pathomechanisms including RNA toxicity and DPR proteins	[94,129,138]
*SOD1*	21q22.11	1993	Antioxidant enzyme	15–20%	1–2%	Over 200 mutations identified; most studied ALS gene; first successful gene therapy target (Tofersen)	[93,139,140]
*TARDBP (TDP-43)*	1p36.22	2008	RNA processing/metabolism	4–5%	1–2%	TDP-43 pathology present in ~97% of all ALS cases; primarily missense mutations in C-terminal domain	[136]
*FUS*	16p11.2	2009	RNA processing/metabolism	4–5%	<1%	Associated with early-onset and aggressive disease course; primarily C-terminal mutations affecting nuclear localization	[141]
*OPTN*	10p13	2010	Vesicular transport	<4%	<0.4%	OPTN defects result in mitophagy disorder, protein aggregation, neuroinflammation, vesicular transport, neuronal axonal degeneration, oxidative stress	[142]
*p62/SQSTM1*	5q35.3	2011	Autophagy	2–3%	<1%	Depletion of p62 protein levels inhibits LC3 recruitment to autophagosomes and has been shown to increase cell death induced by mutant huntingtin	[143]
*ATXN2*	12q24.12	2010	RNA processing	1–2%	<1%	Intermediate-length polyQ expansions (27–33 repeats) increase ALS risk; longer expansions cause SCA2	[144]
*VCP*	9p13.3	2010	Protein degradation, autophagy	1–2%	<1%	Also associated with inclusion body myopathy and frontotemporal dementia (IBMPFD)	[145]
*UBQLN2*	Xp11.21	2011	Protein degradation	1–2%	<1%	X-linked dominant inheritance; affects protein homeostasis; associated with ALS-FTD	[146]
*PFN1*	17p13.2	2012	Cytoskeleton dynamics	1–2%	<1%	Mutant PFN1 contributes to ALS pathogenesis by altering actin dynamics and inhibiting axon outgrowth	[147,148]
*TBK1*	12q14.2	2015	Autophagy, inflammation	1–2%	<1%	Haploinsufficiency mechanism; involved in autophagy and inflammatory pathways	[149]
*NEK1*	4q33	2016	DNA damage response, axonal growth	1–2%	1–2%	Loss-of-function variants; identified through large-scale genome-wide association studies	[150]

## 5. Conclusions and Future Directions

In this review, we sought to critically analyze and synthesize current evidence on how three key innovative approaches are transforming ALS research and clinical practice: emerging biomarker discovery for early diagnosis and patient stratification, multi-omics integration for understanding disease heterogeneity and identifying therapeutic targets, and advanced gene therapy strategies for disease-modifying interventions.

The landscape of ALS research is undergoing a fundamental transformation as these approaches converge to reshape our understanding and treatment of this devastating disease. Our analysis demonstrates that ALS research is transitioning toward mechanism-driven, personalized medicine strategies through three key paradigm shifts: the evolution from single biomarkers to integrated multi-modal diagnostic panels; the transition from single-omics studies to comprehensive platforms that reveal distinct molecular endotypes; and the advancement from symptomatic treatments to precision gene therapies targeting specific genetic causes.

Looking toward the future, several critical priorities emerge that will determine the success of this transformation. In biomarker development, the immediate focus must be on standardizing multi-modal diagnostic panels that seamlessly integrate fluid-based markers like neurofilament light chain and exosome-derived miRNAs with advanced neuroimaging and electrophysiological assessments. This integration requires establishing unified protocols, validation frameworks, and regulatory pathways that can support clinical implementation across diverse healthcare settings. For multi-omics integration, the path forward demands developing computational platforms capable of translating complex molecular signatures into clinically actionable insights. This includes creating standardized data integration pipelines, establishing large-scale biobanks with longitudinal patient samples, and developing artificial intelligence tools that can predict treatment responses based on individual molecular profiles. Critical infrastructure investments are needed in bioinformatics capabilities and collaborative data-sharing platforms. Gene therapy advancement requires a multi-pronged approach focusing on combinatorial strategies that can simultaneously target multiple pathogenic pathways while addressing current limitations in delivery efficiency and therapeutic durability. Priority areas include developing next-generation delivery vehicles that can overcome blood–brain barrier restrictions, creating modular gene therapy platforms that can be adapted to different genetic subtypes, and establishing manufacturing capabilities that can support personalized genetic interventions at scale.

Precision medicine in ALS will require unprecedented collaboration across disciplines, sustained investment in translational research infrastructure, and the development of adaptive regulatory frameworks capable of accommodating the complexity of individualized therapeutic approaches. While significant technical, regulatory, and economic challenges remain, the convergence of these innovative approaches provides genuine cause for optimism in transforming ALS from an invariably fatal condition toward a manageable disorder with personalized treatment strategies.

## Figures and Tables

**Figure 1 ijms-26-05671-f001:**
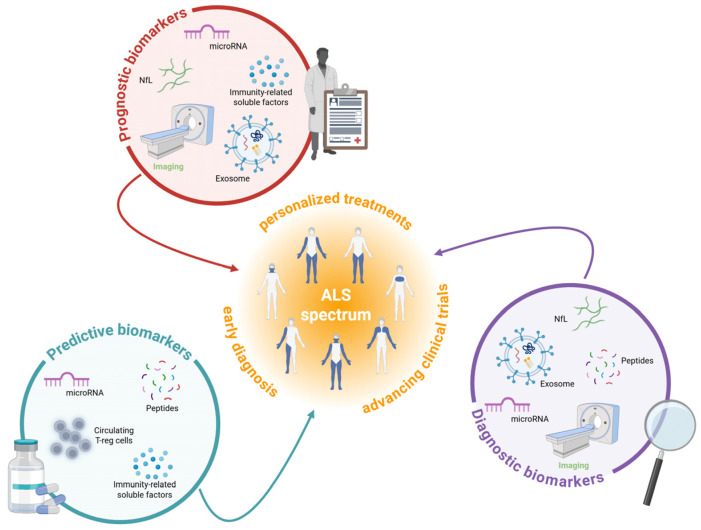
An overview of the main categories of biomarkers for ALS. The figure shows fluid (black) and non-fluid biomarkers (green) investigated in association with the ALS spectrum. Each one can serve as a diagnostic, prognostic, and/or predictive tool, contributing to improve patients’ early diagnosis and advancing clinical trials, hence developing personalized treatments. Abbreviations: ALS, amyotrophic lateral sclerosis; NfL, neurofilament light chain Created in BioRender. Marcuzzo, S. (2025) https://BioRender.com/ahlgwgm.

**Figure 2 ijms-26-05671-f002:**
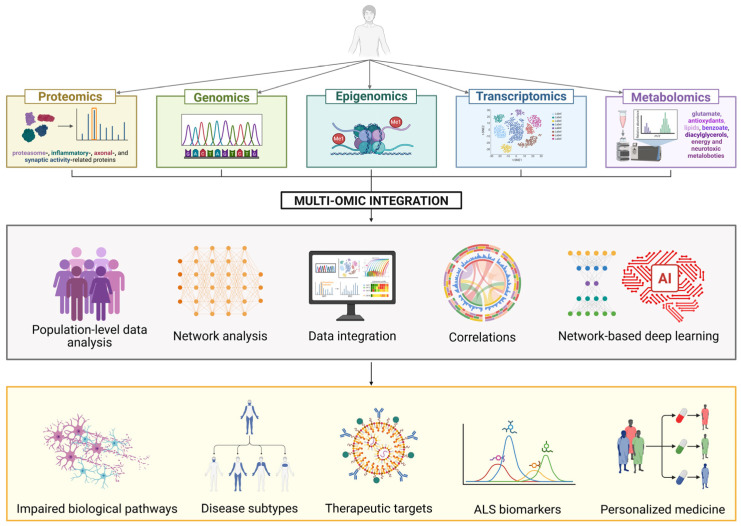
An overview of the multi-omics integration approach applied to ALS. The figure illustrates how different omics layers, such as proteomics, genomics, epigenomics, transcriptomics, and metabolomics, can be simultaneously analyzed through an integrated approach. This method encompasses population-level data analysis, molecular network construction and interpretation, correlation mapping, and network-based deep learning techniques. The ultimate goal of multi-omics integration is to uncover the molecular mechanisms underlying ALS pathophysiology and identify novel therapeutic targets for innovative treatments. Furthermore, given the heterogeneous nature of ALS, the discovery of disease-specific biomarkers could enable molecular-based patient stratification, which is a critical step toward the development of personalized medicine. Created in BioRender. Marcuzzo, S. (2025) https://BioRender.com/ahlgwgm.

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
