# Peer review of "Perspectives in Amyotrophic Lateral Sclerosis: Biomarkers, Omics, and Gene Therapy Informing Disease and Treatment"

_ijms, 2025, doi:10.3390/ijms26125671_

Round 1

Reviewer 1 Report

Comments and Suggestions for Authors

Dear researchers,

Congratulations for the work, below, I send my observations and comments to the paper:

The introduction presents a clear line about ALS, presenting key or guiding findings for treatment. However, the title you present says “Innovative Perspectives in Amyotrophic Lateral Sclerosis: Emerging Technologies and Future Directions”, then, I have the following doubts:

1.- What is the starting point that allows this presentation?

2.- In a clear way, is there a before and an after considering Emerging Technologies in comparison to the already existing ones or the ones that were used?

3.- The research objective is broad, however, it is not clear.

Regarding this last point it is ample, and the presentation of the results should be clearly evidenced in the total of the document.

4.- What do the existing systematic or bibliometric reviews on the subject say, or what are the directions of the existing systematic or bibliometric reviews, what is the originality of your research?

5.- Methodologically, reviews can be presented in different ways, scoping, bibliometric, systematic, narrative, literature, bibliographic, among others. All of them present a clear methodology to take into account where the information was obtained, as well as the use of the databases that have been used, research questions, objectives or hypotheses, but I believe that this work should be methodologically supported by other proposals, perhaps some others already published in the journal. In simple words, why did they arrive at this methodology and not another?

6.- Review the use of citations and references. There are 150 references, but, is there any comparative way in a table to offer a reading considering the proposed objectives?

Author Response

Referee # 1:

Congratulations for the work, below, I send my observations and comments to the paper:

The introduction presents a clear line about ALS, presenting key or guiding findings for treatment. However, the title you present says “Innovative Perspectives in Amyotrophic Lateral Sclerosis: Emerging Technologies and Future Directions”, then, I have the following doubts:

1.- What is the starting point that allows this presentation?

2.- In a clear way, is there a before and an after considering Emerging Technologies in comparison to the already existing ones or the ones that were used?

We thank the Reviewer for this insightful and constructive comment. In response to your questions, we have revised the text to clearly define the starting point of our discussion. The distinction between past, present and future approaches is now more explicitly emphasized throughout the text, specifically in the introduction section (lines 56-74, 77-93), in section 2 (lines 97-103, 154-156, 177-179, 194-198, 237-239, 326-336), 3 (338-342, 467-473, 538-541), and 4 (839-849) in Table 1, in order to better guide the reader through the evolution of the field. Based on your comment, we have also carefully revised the title to better align the title with the overall content and purpose of the review The new title is the follows “Perspectives in Amyotrophic Lateral Sclerosis: Biomarkers, Omics and Gene Therapy Informing Disease and Treatment”.

We believe that these changes help improve the clarity and coherence of the manuscript.

3.- The research objective is broad, however, it is not clear. Regarding this last point it is ample, and the presentation of the results should be clearly evidenced in the total of the document.

We thank the reviewer for this constructive observation, which has helped us significantly improve the manuscript's clarity and structure. In response to this comment, we have clarified the research objective by adding a comprehensive objective statement at the end of the Introduction section (lines 81-94) that clearly defines the three key innovative approaches examined in this review: biomarker discovery, multi-omics integration, and gene therapy development. Accordingly, we have also carefully revised the title of the review to better align the title with the overall content and purpose of the review. The new title is “Perspectives in Amyotrophic Lateral Sclerosis: Biomarkers, Omics and Gene Therapy Informing Disease and Treatment

Besides, we have enhanced the presentation of key findings throughout the document by adding a short summary of the key findings at the end of each major section (sections 2, 3, and 4) that synthesize the main evidence presented and highlight the current state of knowledge.

4.- What do the existing systematic or bibliometric reviews on the subject say, or what are the directions of the existing systematic or bibliometric reviews, what is the originality of your research?

Thank you for your important question. Existing systematic and bibliometric reviews on ALS often focus on specific domains, such as genetic mutations, clinical trials, or therapeutic modalities, analyzing trends and efficacy of treatments through structured, data-driven approaches. However, few of them provide an integrated perspective that connects biomarkers, multi-omics approaches, and emerging gene therapies within a single framework.

The originality of our research lies in providing an integrative narrative framework that bridges three traditionally separate research domains, biomarker discovery, multi-omics integration, and gene therapy development, within a unified translational perspective. Our work emphasizes how multi-omics data and biomarker discovery are converging with gene therapy technologies, highlighting their synergistic potential to enable precision medicine approaches in ALS.

In this sense, our review complements and expands on the scope of existing systematic or bibliometric studies, offering a broader conceptual and translational overview that may inform future research directions and clinical.

For clarity, we have strengthened this integrative perspective by adding explanatory statements in the introduction (lines 81-94) and conclusions (lines 858-897) that explicitly position our work within the broader landscape of ALS research reviews.

5.- Methodologically, reviews can be presented in different ways, scoping, bibliometric, systematic, narrative, literature, bibliographic, among others. All of them present a clear methodology to take into account where the information was obtained, as well as the use of the databases that have been used, research questions, objectives or hypotheses, but I believe that this work should be methodologically supported by other proposals, perhaps some others already published in the journal. In simple words, why did they arrive at this methodology and not another?

Thank you for your valuable comment. In this work, we employed a narrative review approach, integrating both original research articles and review papers. This methodological choice was guided by our aim to offer a comprehensive and up-to-date overview of key advances in ALS, particularly focusing on biomarkers, multi-omics integration, and gene therapy strategies.

Given the broad and evolving nature of the topic, as well as the heterogeneity of the included sources, we found that a narrative approach allowed greater flexibility in synthesizing and interpreting complex findings from diverse study types. We also took into account methodological standards from similar reviews published in this journal and in the wider literature, ensuring that our selection of databases, search strategies, and inclusion criteria were transparent and appropriate for the objectives of our study.

6.- Review the use of citations and references. There are 150 references, but, is there any comparative way in a table to offer a reading considering the proposed objectives?

We thank the Reviewer for this valuable comment, which allowed us to revise the manuscript and include a table (Table 1) outlining both traditional and emerging diagnostic, prognostic, and predictive methods in ALS.

Reviewer 2 Report

Comments and Suggestions for Authors

The manuscript is a review that explores innovative approaches in the diagnosis and treatment of amyotrophic lateral sclerosis (ALS). The authors analyze: New biomarkers (such as neurofilaments, miRNAs, exosomes) for early diagnosis and patient stratification; Advanced imaging and neurophysiology techniques; Multi-omics approaches (genomics, proteomics, metabolomics, epigenomics) to better understand the complexity of the disease; Emerging therapies such as antisense oligonucleotides (ASOs), RNA interference, CRISPR/Cas9, and new delivery strategies across the blood-brain barrier.

The goal is to highlight how the integration of molecular neuroscience, biomedical engineering, and personalized medicine can lead to earlier diagnoses and truly disease-modifying treatments.

The manuscript is well-structured and rich in information, but there are several aspects the authors could improve to enhance clarity, impact, and scientific rigor. Here is a summary of the main areas for improvement:

  • Some paragraphs are extremely dense and include a high volume of technical information that risks getting lost. The authors could:

    • Use shorter and more focused subsections.

    • Include summary diagrams or tables to compare approaches (e.g., type of biomarker, fluids analyzed, specificity, sensitivity).

  • The figures are useful but not thoroughly discussed in the text. The authors should better explain in the body of the manuscript what they show and what added value they provide.

  • The "Conclusions and Future Directions" section is optimistic but too general.

Author Response

 Referee # 2:

The manuscript is a review that explores innovative approaches in the diagnosis and treatment of amyotrophic lateral sclerosis (ALS). The authors analyze: New biomarkers (such as neurofilaments, miRNAs, exosomes) for early diagnosis and patient stratification; Advanced imaging and neurophysiology techniques; Multi-omics approaches (genomics, proteomics, metabolomics, epigenomics) to better understand the complexity of the disease; Emerging therapies such as antisense oligonucleotides (ASOs), RNA interference, CRISPR/Cas9, and new delivery strategies across the blood-brain barrier.

The goal is to highlight how the integration of molecular neuroscience, biomedical engineering, and personalized medicine can lead to earlier diagnoses and truly disease-modifying treatments.

The manuscript is well-structured and rich in information, but there are several aspects the authors could improve to enhance clarity, impact, and scientific rigor. Here is a summary of the main areas for improvement:

Some paragraphs are extremely dense and include a high volume of technical information that risks getting lost. The authors could:

Use shorter and more focused subsections.

We thank the Reviewer for the insightful and constructive comments. We appreciate your positive assessment of the manuscript’s structure and content, as well as your valuable suggestions for improvement. We have revised the text by introducing shorter and more focused subsections specifically in the section 2 and 3.

Include summary diagrams or tables to compare approaches (e.g., type of biomarker, fluids analyzed, specificity, sensitivity).

We have added a new Table 1 to the revised manuscript, which provides an overview of both traditional and emerging diagnostic, prognostic, and predictive approaches in ALS.

The figures are useful but not thoroughly discussed in the text. The authors should better explain in the body of the manuscript what they show and what added value they provide.

We have addressed this point by expanding the discussion of the Figures in the main text (section 2 lines 112-124; section 3 lines 353-356), clarifying what they illustrate and highlighting the added value they provide to the manuscript.

The "Conclusions and Future Directions" section is optimistic but too general.  

In response to this feedback, we have comprehensively rewritten the section "Conclusions and Future Directions", that is now more specific and actionable.

Reviewer 3 Report

Comments and Suggestions for Authors

The review is nicely written with a clear progression of the problems and suggested solutions. Limitations and arguments were also provided by the authors. Minor comments are suggested:

  • Refer to the fluid- and non-fluid-based techniques in the abstract.
  • Lines 16-17: Write abbreviations in full.
  • Line 25: CRISPR/Cas9 gene editing systems.
  • Line 35: Define "bulbar onset."
  • Line 47: Write "FUS" in full.
  • Lines 81-90: Revise the use of numbers; you may instead start with a general statement and then continue with more specific ones.
  • Lines 113-123: This part could be shortened.
  • Line 151: ... that could serve.
  • Line 181: Define Tregs in terms of molecular (flow cytometric) identity. 
  • Line 195: ... immunohistochemistry-based.
  • Lines 218-223: Add specific details.
  • Line 229: Don't start a paragraph with an abbreviation.
  • The distinction between fluid- and non-fluid-based techniques is not clear on the figure. Authors may label with different colors and add a brief explanation to the figure caption.
  • Line 287: Transcriptomic studies play ...
  • Figure 2 lists general headlines. Specific examples for the tested proteins and metabolites can be added.
  • The conclusion should focus on future implications of the summarized work and propose novel research directions for ALS patients. 

Author Response

Referee # 3:

The review is nicely written with a clear progression of the problems and suggested solutions. Limitations and arguments were also provided by the authors.

We sincerely thank the Reviewer for this positive assessment of our work.

Minor comments are suggested:

Refer to the fluid- and non-fluid-based techniques in the abstract.

We have modified the Abstract accordingly.

Lines 16-17: Write abbreviations in full.

According to this comment, abbreviations are now reported in full upon their first appearance in the Abstract.

Line 25: CRISPR/Cas9 gene editing systems.

Line 35: Define "bulbar onset."

Line 47: Write "FUS" in full.

Line 151: ... that could serve.

Line 195: ... immunohistochemistry-based.

Line 229: Don't start a paragraph with an abbreviation.

Line 287: Transcriptomic studies play ...

We thank the Reviewer for these detailed editorial suggestions. We have modified the text accordingly to address all the specified points.

Lines 81-90: Revise the use of numbers; you may instead start with a general statement and then continue with more specific ones.

According to this reviewer‘s comment, we have modified the text (please refer to page 4, lines 113-123).

Lines 113-123: This part could be shortened.

We thank the Reviewer for this suggestion. Having restructured the text and balanced the content across all sections, we believe that the information provided in this section is appropriate and that the level of detail is consistent with the treatment of other fluid biomarkers throughout the manuscript.

Line 181: Define Tregs in terms of molecular (flow cytometric) identity. 

We thank the reviewer for this important suggestion. We have modified the text to include the molecular definition of Tregs by adding their flow cytometric identity as FOXP3+ and CD25+ T cells, which provides readers with the specific molecular markers used for their identification and characterization in research and clinical applications.

Lines 218-223: Add specific details.

We revised the text and add the details (lines 293-297).

The distinction between fluid- and non-fluid-based techniques is not clear on the figure. Authors may label with different colors and add a brief explanation to the figure caption.

We thank the reviewer for this comment. We now revised the figure 1 and its legend.

Figure 2 lists general headlines. Specific examples for the tested proteins and metabolites can be added.

We thank the reviewer for this suggestion. We now revised the figure 2 and add tested proteins and metabolites

The conclusion should focus on future implications of the summarized work and propose novel research directions for ALS patients. 

We have comprehensively rewritten the section "Conclusions and Future Directions", that is now more specific and actionable.